# Bladder Reconstruction in Cats Using In-Body Tissue Architecture (iBTA)-Induced Biosheet

**DOI:** 10.3390/bioengineering11060615

**Published:** 2024-06-16

**Authors:** Naoki Fujita, Fumi Sugiyama, Masaya Tsuboi, Hazel Kay Nakamura, Ryohei Nishimura, Yasuhide Nakayama, Atsushi Fujita

**Affiliations:** 1Laboratory of Veterinary Surgery, Graduate School of Agriculture and Life Sciences, The University of Tokyo, Tokyo 113-0032, Japan; 2Laboratory of Veterinary Pathology, Graduate School of Agriculture and Life Sciences, The University of Tokyo, Tokyo 113-0032, Japan; 3Osaka Laboratory, Biotube Co., Ltd., Osaka 565-0842, Japan; y.nakayama@biotube.co.jp

**Keywords:** in body tissue architecture (iBTA), cat, urinary tract reconstruction

## Abstract

Urinary tract diseases are common in cats, and often require surgical reconstruction. Here, to explore the possibility of urinary tract reconstruction in cats using in-body tissue architecture (iBTA), biosheets fabricated using iBTA technology were implanted into the feline bladder and the regeneration process was histologically evaluated. The biosheets were prepared by embedding molds into the dorsal subcutaneous pouches of six cats for 2 months. A section of the bladder wall was removed, and the biosheets were sutured to the excision site. After 1 and 3 months of implantation, the biosheets were harvested and evaluated histologically. Implantable biosheets were formed with a success rate of 67%. There were no major complications following implantation, including tissue rejection, severe inflammation, or infection. Urinary incontinence was also not observed. Histological evaluation revealed the bladder lumen was almost entirely covered by urothelium after 1 month, with myofibroblast infiltration into the biosheets. After 3 months, the urothelium became multilayered, and mature myocytes and nerve fibers were observed at the implantation site. In conclusion, this study showed that tissue reconstruction using iBTA can be applied to cats, and that biosheets have the potential to be useful in both the structural and functional regeneration of the feline urinary tract.

## 1. Introduction

Ureteral obstruction is a common cause of urinary dysfunction in cats, with increasing incidence in recent years. A major cause of feline urinary obstruction is urolithiasis, with calcium oxalate uroliths accounting for 98% of cases [1,2]. Urinary obstruction due to urolithiasis often results in inflammation and dilation of the ureter which progresses from the obstruction site, and is a potentially life-threatening condition requiring immediate relief of the obstruction. Due to their narrow ureters, many cats are forced to undergo surgical treatment, such as ureterotomy, ureteronephrectomy, and neoureterocystotomy. However, the clinical outcome after surgery is not always favorable, and various postoperative complications such as persistent or recurrent ureteral obstruction and urine leakage are known to occur [2,3,4]. For cases that suffer irreversible dysfunction following surgery, surgical reconstruction of the ureter is necessary.

To date, various materials have been investigated as substitutes for urinary tract reconstruction. The patient’s own intestine has been the most commonly used material for urinary reconstruction in humans [5]. However, surgical complications resulting from the use of gastrointestinal tissue, such as infection, urinary leakage, stone formation, and intestinal obstruction, have been described [6,7]. Therefore, many alternative approaches have been developed as practical and functional substitutes for natural bladder tissue. Synthetic materials such as polyglycolic acid (PGA), poly(ε-caprolactone) (PCL), and polytetrafluoroethylene (PTFE) are reasonable candidates with certain strengths and suitable microstructures [8,9]. However, these synthetic materials can also induce a chronic inflammatory response, leading to urinary tract infections and bladder necrosis [8,10]. Collagen-based scaffolds, such as acellularized small intestinal submucosa (SIS), have also been used for bladder and ureteral reconstruction [11,12,13,14]. These scaffolds are advantageous in that not only their structure but also their biological activities closely resemble those of the natural bladder [15]. However, the physiological environment can be altered by decellularization [16,17] and possible residual immunogens that can remain even after sterilization can trigger an immune response [18,19,20].

Recently, Nakayama et al. developed a novel in vivo tissue engineering technology, called in-body tissue architecture (iBTA), which is based on the tissue-encapsulation phenomenon of foreign material in living bodies [21,22]. A mold consisting of a silicone tube with a stainless-steel rod placed inside, enclosed in a cylindrical plastic cage with linear windows, is first embedded into the subcutaneous tissue. After a few months, collagenous connective tissue forms around the silicone tube [23]. The resulting tubular tissue, called a “biotube”, consists of an abundant extracellular matrix, and its utility as a graft for the reconstruction of collapsed blood vessels has been evaluated for human patients suffering from cardiovascular and angiopathic diseases [24,25,26]. Watanabe et al. also reported that, in rabbits, a biotube implanted into the carotid artery successfully functioned as a scaffold for cells, including endothelial cells, fibroblast cells, and myocytes. After 6 months of implantation, reconstruction of the layered structure of the blood vessel wall, including the endothelial layer, submucosal layer, smooth muscle layer, and serosa, was achieved [24]. Grafts engineered using iBTA technology have also been reported to have the following advantages: no immunological rejection (autologously engineered grafts), no toxicity, high biocompatibility, as well as being able to adapt to the growth of the recipient [21]. Moreover, the grafts can be engineered into a wide range of shapes and sizes to suit the recipient by adjusting the shape and size of the molds. Most importantly, this technique does not require expensive or time-consuming procedures such as complex cell management or exceptionally clean laboratory facilities. Based on these results, biotube implantation has recently been applied to hemodialysis patients for the reconstruction of collapsed shunt vessels [27].

The wall of the urinary tract has a layered structure, consisting of a urothelium layer at the luminal surface, a submucosal layer, and a smooth muscle layer. This layered structure closely resembles that of blood vessels. The structural similarity between blood vessels and the urinary tract suggests the possibility that iBTA can also be applied to not only the vascular system but also to the urinary tract, and offers a promising method for ureteral reconstruction. Another study using dogs demonstrated that bladder wall reconstruction could be achieved by transplanting autologous collagen connective tissue membranes (biosheets) fabricated using iBTA technology, suggesting that biosheets also have the potential to reconstruct the urinary tract [28]. However, so far, there have been no reports of graft fabrication, such as biotubes and biosheets, using iBTA technology in cats, or the utilization of these grafts in the reconstruction of the feline urinary tract. In this study, we fabricated biosheets using iBTA technology in cats and implanted the biosheets into the urinary bladder in order to investigate their biocompatibility and utility as a scaffold for the reconstruction of the feline urinary tract.

## 2. Materials and Methods

### 2.1. Animals

Six healthy domestic cats (two males and four females) aged 5 months with a body weight of 2–3 kg were used in this study. Cats were separately housed in a stainless-steel cage during this study in a controlled environment (12 h/12 h light–dark cycle, temperature 18–28 °C) throughout the course of the study. All cats received a prescription diet (pH Control 1, Royal Canin, Aimargues, France) to minimize the risk of urolithiasis, and were allowed free access to water. All animal experiments were approved by the Animal Care Committee of the Graduate School of Agricultural and Life Sciences at the University of Tokyo. Table 1 shows the individual characterization of the cats used in this study.

### 2.2. Mold Preparation for Graft Fabrication

The mold was assembled from two parts, consisting of a silicone tube with a stainless-steel rod inserted inside, enclosed in a cylindrical plastic cage with varying types of linear windows (Figure 1A). The assembly was fixed with a flat plastic cap at both ends. Each of the components was constructed using a 3-D rapid prototyping printer (Project 3510 HD, 3D Systems, Rock Hill, SC, USA). The gap between the silicone tube and outer plastic cage was set at 0.5–1 mm. The size of the mold was 5.5 cm in length and 1.0–1.5 cm in diameter.

### 2.3. Embedding of Mold and Graft (Biosheet) Fabrication

All surgical procedures were performed by two surgeons (A. F. and N. F.) under general anesthesia. Atropine sulfate (0.01 mg/kg, IV), butorphanol (0.2 mg/kg, IV), and ampicillin (30 mg/kg, IV) were administered as preanesthetic medications. General anesthesia was induced with alfaxalone (5 mg/kg, IV) and maintained with isoflurane (2.0%). Two molds were separately embedded into the subcutaneous tissue of the right or left lumbar region in each cat. The skin incision was closed with a 4-0 nylon suture. Ampicillin (30 mg/kg, IV, BID) and robenacoxib (2 mg/kg, SC, SID) were administered postoperatively for 3 days. The molds were harvested after 2 months under general anesthesia as described above. By trimming the surrounding tissue and removing the outer plastic cage, a tubular tissue structure was obtained. After confirming there was no defect in the tissue graft, the tubular structure was cut longitudinally to obtain a sheet-shaped graft (biosheet). The biosheet was then trimmed to a size of 1 × 2 cm and preserved in 70% ethanol. A piece of residual tissue was fixed in 10% formalin and embedded in paraffin for histopathological examination (hematoxylin–eosin (HE) and Masson’s trichrome (MT) stains).

### 2.4. Biosheet Implantation

Under general anesthesia, a section of the cranial ventral bladder wall (1 × 2 cm) was excised. The biosheet was washed in saline before implantation, placed into the excised site, and then sutured using a simple continuous suture pattern with a 4-0 nylon suture. After implantation, the bladder was filled with saline to test for leakage and covered with omentum. Finally, the abdominal wall and skin incision were closed with a 4-0 PDS and a 4-0 nylon suture, respectively. A urinary catheter was inserted into the bladder. To minimize the suffering and distress of the animals, fentanyl (3–5 μg/kg/h) and fluid (Fisio140^®^: Otsuka Co., Tokyo, Japan) were intravenously administered during the preoperative period.

### 2.5. Clinical Evaluation after Implantation

The animals were hospitalized for 5 days after implantation and received careful monitoring for signs of hematuria, pollakiuria, and urine leakage. Animals were also administered ampicillin (30 mg/kg, IV) once daily, and robenacoxib (2 mg/kg, SC) twice daily for 1 week. The urinary catheter was removed 3 days after implantation. Serum biochemical analysis including blood urea nitrogen (BUN) and creatinine concentrations were closely monitored during hospitalization. Ultrasound examination of the implantation site was also performed at 1 day, 1 month, and 3 months post implantation.

### 2.6. Histological Evaluation

Three of the six cats were randomly selected and humanely euthanatized after 1 and 3 months of implantation. The bladders were fixed with 10% formalin for 24 h and tissue containing the implanted biosheets were embedded in paraffin. Sections of 4 μm thickness were then processed and stained by HE and MT. Immunohistochemistry was also performed using monoclonal antibodies against α-smooth muscle actin, Desmin, CD31, and S-100 to evaluate the migration and maturation of myofibroblasts, myocytes, vascular endothelium, and neural fibers, respectively.

## 3. Results

### 3.1. Biosheet Characterization

After 2 months of implantation in the subcutaneous tissue, the mold was encapsulated with collagenous connective tissue (Figure 1B). A tubular tissue structure was obtained from the silicone tube after careful trimming of the surrounding connective tissue (Figure 1C) and a sheet-like structure was obtained after longitudinal incision (Figure 1D). However, 4 of the 12 molds resulted in incomplete tissue grafts with thin or defective walls. The remaining eight grafts were smooth and whitish in color, with a thickness of 0.5–1 mm according to the designated mold. One cat received a biosheet harvested from another cat due to incomplete fabrication of the autologous biosheet. HE and MT staining revealed that the tissues obtained were fibrous and rich in collagen, with fibroblasts infiltrating into the connective tissue (Figure 1E,F).

### 3.2. Biosheet Implantation and Postoperative Evaluation

The biosheets were able to tolerate surgical manipulation and could be easily sutured to the bladder wall (Figure 2A,B). No urine leakage was observed perioperatively. Hematuria was observed soon after surgery in all cats, but resolved within 3 days, after which the urinary catheter was removed. Serum biochemical analysis including BUN and creatinine did not indicate any severe abnormalities in any of the cats. Based on the ultrasound findings, urine leakage was not observed throughout the observation period. Ultrasound examination of the region containing the biosheet at 1 month post implantation showed a thickened bladder wall with an irregular margin (Figure 3A). However, these abnormal findings disappeared after 3 months, and the implanted area appeared to be as smooth as a native bladder (Figure 3B).

### 3.3. Histological Examination

There was severe adhesion of the implantation site to the omentum, and the biosheets could not be identified by gross observation (Figure 3C,D). The implantation site was stiff and seemed to have little flexibility for 1 month after implantation. The site regained a flexibility similar to that of a native bladder wall after 3 months.

HE and MT staining revealed hyperproliferation of the urothelium by which the lumen of the bladder was almost entirely covered (Figure 4A,B). There was observed remarkable ingrowth of fibroblastic cells into the biosheets under the urothelium layer (Figure 4C). Immunohistochemistry revealed these fibroblastic cells were positive for αSMA, indicating that they were myofibroblasts (Figure 4D). However, cells expressing Desmin, a marker for mature myocytes, were not detected (Figure 4E). On the other hand, CD31-positive cells were observed in this area, indicating micro-neovascularization 1 month after implantation (Figure 4F). No NF200-positive nerve fibers were detected (Figure 4G).

After 3 months, the implantation site had a smooth lumen and was completely covered by urothelial cells forming a pseudostratified-like urothelium (Figure 4H). MT staining revealed a well-organized extracellular matrix (Figure 4I). A bundle of mature cells in the implanted region was observed and these cells were positive not only for αSMA but also for Desmin (Figure 4J–L). These findings indicate the processes involved in reconstructing the submucosal layer and the smooth muscle layer to resemble the layer structure of the natural bladder wall (Figure A1). Further, there was development of vasculature with CD31 expression in this region (Figure 4M). Moreover, NF200-positive spindle-shaped cells were observed between the bundles of smooth muscle cells (Figure 4N).

## 4. Discussion

The fabrication of tissue grafts such as biotubes and biosheets using iBTA technology has been reported in rats rabbits dogs, goats, and humans [21,25,28,29,30]. In this study, we demonstrated that it is also possible to make such grafts in cats. However, the success rate for the formation of implantable biosheets was 67% (8/12), which was lower than the previous studies in goats [31]. Although inflammation due to infection after mold implantation is reported to be a factor in graft formation failure [28], there were no findings indicating infection in this study. In order to form a graft of sufficient thickness, blood vessel infiltration into the mold and a sufficient supply of nutrients are necessary [31]. Therefore, it may be that graft formation failure occurred as a result of insufficient vascularization within the mold. It is also possible that the immune response to silicone in cats is weaker than in other species. It has been reported that the windows of the outer cylindrical cage of the mold help promote cell infiltration, and that the number and size of these windows are related to the efficiency of graft formation [31]. Therefore, it may be necessary to make adjustments to the mold for cats, such as increasing the number or size of the windows. Further, in this study, the efficiency of graft formation differed between the molds embedded in the right lumbar region and those embedded in the left, suggesting that differences in the rate of graft formation may not be due to biological factors alone. In general, cats tend to rub their bodies against walls or objects as a form of marking behavior. The cats in this study also often rubbed their bodies against their cages, which may have caused mechanical stimulation to the implantation site. These behavioral characteristics of cats may have resulted in an unfavorable environment for graft formation.

Tissue grafts for implantation into the urinary tract need to have high biocompatibility as well as be elastic and flexible enough to store and transport urine [32,33]. The biosheet used for implantation had a thickness of 0.5–1.0 mm and was strong enough to withstand surgical manipulation. No urine leakage was observed immediately after implantation, and the biosheet successfully compensated for the defect in the bladder wall. Therefore, the biosheet was considered to have sufficient biocompatibility to be used as a graft for urinary tract reconstruction. The luminal surface of the bladder, including the implanted biosheet, was mostly covered by urothelium 1 month after implantation and was completely covered with multilayered urothelium after 3 months. A similar process of epithelialization of the luminal surface has also been reported in bladder wall reconstruction studies using PCL and SIS [33,34]. The urothelium plays a role in protecting the bladder wall from urine stimulation. Therefore, early regeneration of the urothelium layer is considered essential for bladder reconstruction [32]. Based on the histological findings in this study, the biosheets were considered to function excellently as a scaffold for the migration of urothelial cells, similar to other synthetic and collagen-based scaffolds. In addition, infiltration of myofibroblasts, which are involved in scar formation, was observed in the biosheet, and it was thought that this contributed to the promotion of epithelialization.

Three months after implantation, reconstruction of the muscle layer was confirmed, along with increased flexibility of the biosheet implantation site. Smooth muscle is essential for bladder compliance and contractility, and its regeneration is induced by adequate angiogenesis and migration of the urothelium [18]. In the areas where muscle cell clusters were observed, a new vascular network had also developed, suggesting that this contributed to the maturation and proliferation of muscle cells resulting in the recovery of contractility. Angiogenesis also promotes nerve fiber infiltration in bladder regeneration [35]. In this study, infiltration of nerve fibers between muscle cells was also observed, suggesting that angiogenesis played an important role in the functional reconstruction of the bladder wall. Although we did not investigate the motility of the bladder smooth muscle, the appearance of nerve fibers suggested the recovery of urine excretory function. Similar results in muscle layer regeneration have been reported in bladder wall reconstruction studies using PCL and SIS. These reports also showed that smooth muscle maturation over the graft material was accompanied by nerve regeneration [10,34]. In the previous study using dogs, the biosheets successfully played a functional role as a scaffold for urothelium regeneration, but no mature muscle cells or nerve fibers appeared even after 3 months of implantation [28]. This may be due to the larger size of the bladder defect (3 × 5 cm) compared to the present study. However, abundant vascularization was also observed, suggesting that smooth muscle and nerve regeneration could be expected with longer observation periods.

The biosheet was shown to function as a scaffold for bladder wall regeneration, similar to other materials, but without any significant complications. Previous reports have shown that using synthetic and collagen-based graft materials can result in complications such as calcification and necrosis of the urinary tract, associated with unfavorable inflammatory and foreign body reactions [8,10,34]. Synthetic materials, such as PGA and PCL, are biodegradable, and their degradation products are relatively strong acids that cause inflammation [36]. Xenogeneic graft materials, such as SIS, are decellularized and sterilized before use; however, the risk of residual immunogens cannot be eliminated entirely. In contrast, the biosheet, being an autologous material, can overcome these risks. Further, no adverse effects were observed in this study, even when using the allogeneic biosheet. It is thought that over 2 months is needed to form an implantable graft using iBTA technology in cats. However, urinary tract obstruction often requires immediate intervention; thus, preparing an autologous transplant would be difficult in clinical practice. In this study, an allogeneic biosheet was implanted in one cat and observed for 3 months. There were no findings suggesting immune rejection, and bladder wall reconstruction was observed similarly to the autologous biosheets. Therefore, allogenic biosheets prepared in advance using iBTA technology could serve as a highly versatile graft material for urinary tract reconstruction, especially for the feline urinary tract. Further, given the absence of any major complications related to the graft material, the biosheet is a promising alternative to conventional methods using synthetic materials and collagen-based scaffolds for urinary tract reconstruction, with potential applications in both veterinary and human medicine.

## 5. Conclusions

Biosheets can be induced by in-body tissue architecture (iBTA) in cats and are expected to have high biocompatibility as graft materials for urinary tract reconstruction, contributing to histological and functional reconstruction.

## Figures and Tables

**Figure 1 bioengineering-11-00615-f001:**
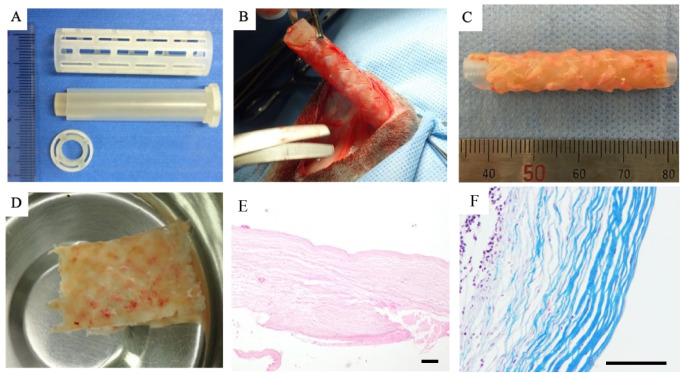
Preparation and implantation of the biosheets: A silicone tube inside a silicone cage was used for fabrication of the biosheets (**A**). After embedding in the dorsal subcutaneous pouch for 2 months, the mold was completely encapsulated in connective tissue (**B**). The tissue around the mold was carefully trimmed and the silicon cage was removed. The space inside the mold was filled with connective tissue (**C**). A biosheet was obtained by cutting longitudinally along the tubular tissue structure (**D**). Hematoxylin and eosin staining (**E**) and Masson’s trichrome staining (**F**) of a biosheet before implantation. The biosheets consisted of abundant collagenous tissue with a few fibroblasts observed. Scale bars: 100 μm.

**Figure 2 bioengineering-11-00615-f002:**
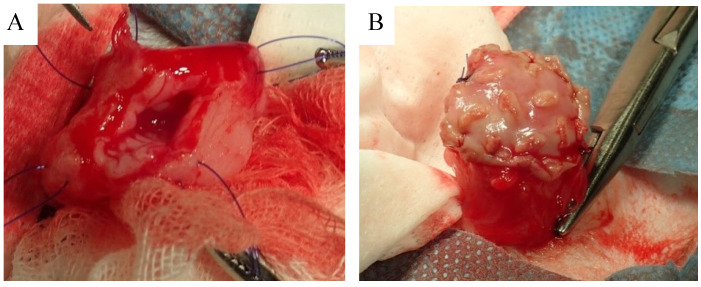
Surgical procedure of biosheet implantation: A 1 × 2 cm-size window was created in the cranial ventral bladder wall (**A**). The biosheet was trimmed to fit the window and then sutured to the bladder wall (**B**).

**Figure 3 bioengineering-11-00615-f003:**
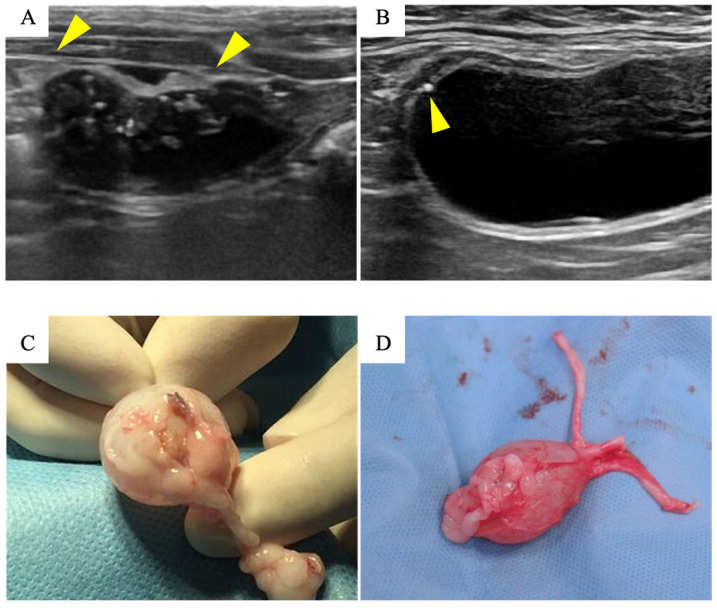
Ultrasound examination and macroscopic findings of the implantation site: Ultrasound of the bladder at 1 month after implantation (**A**). A thickened bladder wall with irregular margins was observed at the implantation site (yellow arrowhead). Ultrasound of the bladder 3 months after implantation (**B**). The thickening resolved and the bladder lumen appeared smooth. The yellow arrowhead indicates the suture. Macroscopic finding 1 month after implantation (**C**). The omentum was strongly adhered to the implantation site, which was stiff and had little flexibility. Macroscopic finding 3 months after implantation (**D**). Although there was still adhesion of the omentum, the implantation site had regained a flexibility similar to that of a native bladder wall.

**Figure 4 bioengineering-11-00615-f004:**
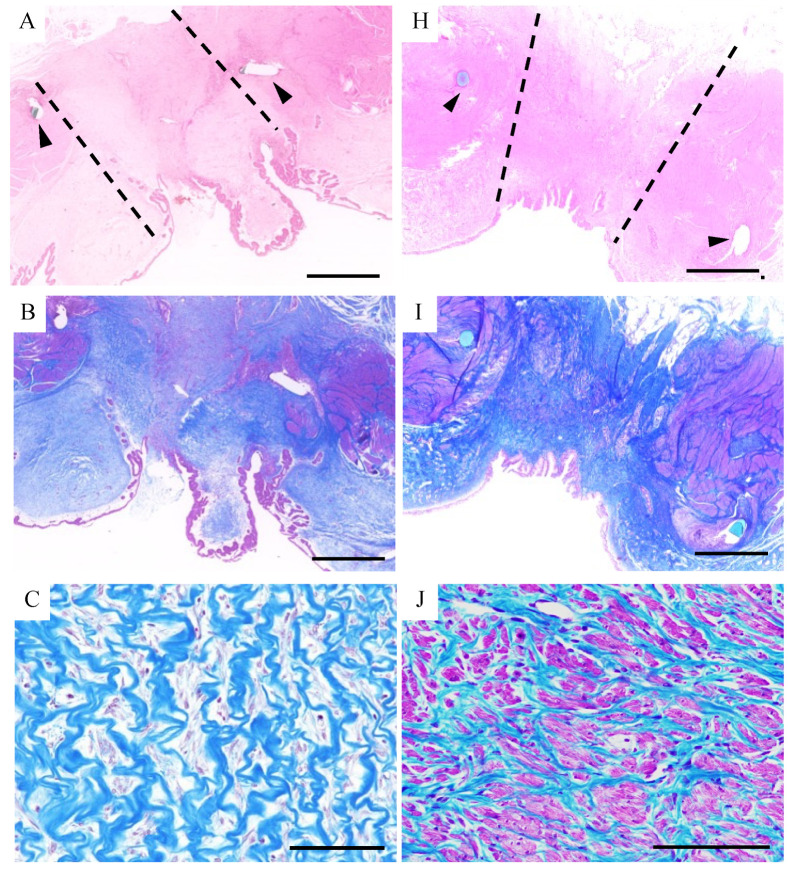
Histological evaluation of the implanted area one month (**A**–**G**) and three months (**H**–**N**) after implantation. The border of the implanted region (dotted line) was detected by the suture remnants (arrow heads) (**A**,**H**). Hematoxylin and eosin staining showed that the luminal surface of the bladder was mostly covered by urothelium 1 month after implantation (**A**) and was completely covered with multilayered urothelium after 3 months (**H**). Masson’s trichrome staining (**B**,**C**,**I**,**J**) shows rough collagenous tissue indicating the biosheet between the native bladder wall, and fibroblasts infiltrating into the collagenous tissue (**B**,**C**). After three months, the layer structure appeared (**I**) and bundles of muscle cells were observed in the implanted region (**J**). At 1 month, there was an abundance of cells positive for αSMA (**D**); however, these cells were negative for Desmin (**E**), suggesting an abundant migration of myofibroblasts. After 3 months, the αSMA-positive cells (**K**) were also positive for Desmin (**L**), indicating the maturation and proliferation of muscle cells. At 1 month post implantation, CD31-positive cells were observed at the implantation site, indicating micro-neovascularization (**F**). There was development of vasculature with CD31 expression at this region at 3 months after implantation (**M**). No nerve fibers were detected by NF200 staining after 1 month (**G**). However, after 3 months, NF200-positive nerve fibers were observed between the regenerated muscle bundles (**N**). Scale bars = 2 cm (**A**,**B**), 1 cm (**H**,**I**), 500 μm (**D**,**E**,**K**,**L**), 100 μm (**C**,**F**,**G**,**J**,**M**,**N**).

**Table 1 bioengineering-11-00615-t001:** Individual characterization of the cats used in this study.

No.	Sex	Year	Body Weight (kg)	Observation Period	Success Rate ^※1^	Implanted Biosheet
1	Male	0.5	2.7	1 month	2/2	autologous
2	Male	0.5	2.4	1 month	2/2	autologous
3	Male	0.5	2.5	1 month	1/2	autologous
4	Female	0.5	2.4	3 months	2/2	autologous
5	Female	0.5	2.4	3 months	1/2	autologous
6	Female	0.5	2.5	3 months	0/2	biosheet from cat#4

^※1^: Two molds were subcutaneously implanted into each cat. The success rate indicates the ratio of formation of biosheets suitable for implantation.

## Data Availability

The original contributions presented in the study are included in the article, further inquiries can be directed to the corresponding author.

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
