# Peer review of "Bladder Reconstruction in Cats Using In-Body Tissue Architecture (iBTA)-Induced Biosheet"

_bioengineering, 2024, doi:10.3390/bioengineering11060615_

Round 1

Reviewer 1 Report

Comments and Suggestions for Authors

Dear All,

Study "Bladder reconstruction in cats using in-body tissue architecture (iBTA)-induced biosheet" by Naoki Fujita et al. deals with the potential use of biosheets fabricated through iBTA technology in order to reconstruct the urinary bladder in cats.

The Introduction section of the manuscript is clearly written presenting urinary tract diseases in cats as a veterinary and research problem, especially how surgical reconstructions are performed for various conditions such as urolithiasis. Through the Introduction, it was presented how iBTA technologies are used in tissue engineering, especially in vascular reconstruction. Thus, the motive and hypothesis on the basis of which the objectives of the study were set are well explained.

In the Materials and Methods section, an overview of the experimental procedure is given, including a description of animals, mold preparation, graft (biosheet) fabrication, biosheet implantation, clinical evaluation of animals, histological evaluation.

However, in this part it is necessary to do the following to improve the manuscript:

1. Along with a brief description of the animals' keeping and caring, let the authors also state what conditions are provided to minimize the suffering and distress of the animals during study.

2. The drawback of the study is that there is no control group for comparison, which would strengthen the scientific validity of the study, so a comparison of the obtained results using iBTA-induced biosheets for bladder reconstruction in cats with other methods of applying tissue engineering, regenerative medicine and reconstructive surgery should be made in the Discussion.

3. Due to the small number of animals in the groups and the heterogeneity of the characteristics, for better monitoring and understanding of the study, I suggest that data be given individually for each animal, e.g. with labels from 1 to 6 to make it easier to characterize - which individual is male and which is female and what is the mass of each of them, and data on clinical condition, date of sacrifice and key findings could also be added.

4. It should state how the animals were selected for euthanasia at one and three months.

5. It can be important information to consider the findings in the study - Did only one surgeon perform one type of procedure or did different veterinary surgeons work on different animals?

The Results section presents the findings of histological and clinical evaluations after implantation, which indicate that the biosheet showed promising biocompatibility and structural integrity, facilitating the regeneration of the layered structure of the bladder wall over time.

In this section you need to do the following:

1. In the description of the images in Figure 4 below the images, the descriptions of the images under A, B, C, H, J and I are missing and it is necessary to add them.

2. In Figure 4, the images under D, E, F, G, K, L, M and N should be larger.

3. In Figure 4, the images under E and G should be clearer.

The Discussion section interprets the results, addressing the factors that influence the success rate of biosheet formation and how this may further influence urinary tract reconstruction. The significant value of the discussion is the comparison with studies on other animal models and consideration of the limitations of the study, as well as the potential clinical applications of its results.

My suggestion is to additionally discuss:

1. the limitations of the study through the heterogeneity of the groups, as well as how perspective is the use of the described technique in veterinary practice on cats and possibly in human medicine.

2. Do as requested according to point 2 in the Materials and Methods review section.

The Conclusions correspond to the previously presented results and their discussion.

Overall Recommendation:

The manuscript provides a valuable insight into the application of iBTA-induced biosheets for bladder reconstruction in cats as progress in the application of tissue engineering in veterinary practice.

But the manuscript could be accepted for publication, after considering the proposed revision.

Reviewer 2 Report

Comments and Suggestions for Authors

iBTA: It should be uniform for in body tissue archtechture through out the manuscript.

Figure should be divided in two, one for graft preparation with full characterization and other for transplantation.

From current explaination it is not clear how this graft was prepared and chararcteried, details should be added.

The development of animal model need full explaination, and its characterization that animal show functional defect.

Comments on the Quality of English Language

Minor changes 

Reviewer 3 Report

Comments and Suggestions for Authors

The aim of the study was to show bladder reconstruction in cats using iBTA. Topic is interesting and study properly performed. However, there are few minimal aspects that Authors should review in order to improve the overall quality of manuscript. 

- Histological comparison between original bladder and iBTA bladder reconstruction should be reported and showed in a specific figure, in order to emphasize similarities. 

- A table reporting clinical features and postoperative outcomes could be reported. 

Round 2

Reviewer 1 Report

Comments and Suggestions for Authors

Dear Sirs, 

Thanks to the authors for respecting and fulfilling the requests and suggestions of the reviewers. Bearing in mind my previous review and the author's responses, I suggest that the manuscript be accepted for publication in its present form.

Reviewer 2 Report

Comments and Suggestions for Authors

Accepted in current form

Comments on the Quality of English Language

Nil